# Relationship between Sucrose Taste Detection Thresholds and Preferences in Children, Adolescents, and Adults

**DOI:** 10.3390/nu12071918

**Published:** 2020-06-29

**Authors:** Sara Petty, Clara Salame, Julie A. Mennella, Marta Yanina Pepino

**Affiliations:** 1Department of Food Science and Human Nutrition, University of Illinois at Urbana-Champaign, Urbana, IL 61801, USA; saraap2@illinois.edu; 2Division of Human Nutrition, University of Illinois at Urbana-Champaign, Urbana, IL 61801, USA; csalame2@illinois.edu; 3Monell Chemical Senses Center, Philadelphia, PA 19104-3308, USA; mennella@monell.org

**Keywords:** sweet taste, detection threshold, preferences, psychophysics, development

## Abstract

To address gaps in knowledge, our objectives were to (1) to determine whether there are age-related changes in sweet taste detection thresholds, as has been observed for sweet taste preferences, and (2) determine whether detection thresholds and taste preferences were significantly related to each other from childhood to adulthood. We combined data from studies that used the same validated psychophysical techniques to measure sucrose taste detection threshold and the most preferred sucrose concentration in children (*n* = 108), adolescents (*n* = 172), and adults (*n* = 205). There were significant effects of age group on both sucrose detection thresholds (*p* < 0.001) and most preferred sucrose concentration (*p* < 0.001). While children had higher sucrose detection thresholds than adolescents, who in turn tended to have higher detection thresholds than adults, both children and adolescent most preferred sucrose concentrations were higher than that of adults (all *p* < 0.05). Among each age group, and when combined, the lowest sucrose concentration detected was not significantly correlated with the most preferred sucrose concentration (all *p* > 0.18). These data provide further evidence that age-related changes in sucrose taste preferences that occur during adolescence cannot be explained by changes in taste sensitivity and that these two dimensions of sweet taste undergo distinct developmental trajectories from childhood to adulthood.

## 1. Introduction

Sweetness, one of the most preferred taste sensations, can be characterized by an array of distinct attributes—some of which can be measured from an early age. In young children, psychophysical studies can provide data relevant to two separate attributes of taste: sensitivity of the system to chemical stimuli and the hedonic valence of the sensation [1]. Measures of sensitivity include taste detection thresholds, just noticeable differences, intensity judgments, and sensory adaptation, whereas measures of hedonics include estimates of pleasantness, liking, or relative preferences. Although sensitivity and hedonics reflect distinct features of the taste system, and this distinction is usually unambiguous, the terms are often used interchangeably, despite little or no evidence that they are directly associated with one another.

Using validated methods that assess the hedonics of the taste system by directly measuring the level of sweetness preferred in participants of varying ages, research has shown that children experience tastes differently than adults, particularly sweet tastes. Compared to adults, children most prefer significantly higher concentrations of nutritive sugars (e.g., sucrose and fructose) [2,3,4,5,6,7,8] and low-calorie sweeteners (e.g., aspartame and sucralose) [2] than adults. For the sweet taste of sucrose, the adult pattern emerges during mid-adolescence [3,5].

To our knowledge, the research suggesting similar age-related changes, beginning in childhood, in the sensitivity of the taste system is limited [9]. While there is a gradual and steady decline in the ability to detect sucrose (i.e., higher concentrations of sucrose are required) from the second to eighth decade [9,10], it is unclear whether such declines begin in childhood [9,11]. There is wide variation in sucrose taste detection thresholds among children [9,11,12,13], and one study reported that 8- to 9-year-old boys had higher sucrose detection thresholds (meaning they were less sensitive) than adults [13]. Such age-related changes in taste sensitivity have been attributed to children’s inability to complete the detection threshold tasks and their shorter attention span [9,11]. However, if sucrose detection thresholds are indeed higher in children than in adults, one explanation why they prefer higher levels of sweet taste is that they are less sensitive to its taste.

In the present study, our objectives were twofold. First, we aimed to determine whether there are age-related changes in taste detection thresholds for sucrose, as there are for sweet taste preferences. Second, we aimed to determine the degree of association between sucrose taste detection thresholds and the most preferred sucrose concentrations, and whether such relationships change from childhood to adulthood. To address these aims, we used psychophysical tools that have been validated for use in pediatric populations (e.g., the NIH Toolbox) [5,14], which allow us to directly measure them in children and not rely solely on maternal reports, and in the case of taste detection thresholds, used clinically [15]. Using identical methods for all participants allowed us to directly compare both sweet taste sensitivity and sweet taste preference across age groups.

## 2. Materials and Methods

### 2.1. Participants

Data from individuals who participated in research studies that included measurements of either sucrose taste detection thresholds alone or both sucrose detection thresholds and the most preferred sucrose concentration were included in this study. These data had never been analyzed for age-related changes in sweet taste sensitivity or how it relates to changes in sweet taste preference over time.

The research was conducted at the Monell Chemical Senses Center in Philadelphia, PA, and Washington University in St. Louis, MO. Participants were grouped based on age, as defined by the World Health Organization [16]: children (<10 years; *n* = 108); adolescents (10–19 years; *n* = 172) and adults (>19 years; *n* = 205). The studies conducted at the Monell Center were comprised of children, adolescents and/or adults. For the studies on adults only, inclusion criteria were women who were either normal weight or with obesity. The studies conducted at Washington University were comprised of adults with obesity only. Inclusion criteria encompassed that participants were healthy with no major medical illness (e.g., diabetes, heart disease, asthma, chronic rhinitis, and food allergies). Adults who were current smokers, and adult women who were pregnant or lactating, were excluded from the analyses because of smoking- or pregnancy-related alterations in chemosensation [17,18]. Race/ethnicity was reported by the adult participants or by the mothers of the children and adolescent participants.

All procedures at the Monell Center were approved by the Office of Regulatory Affairs at the University of Pennsylvania (IRB Review Boards 1 and 7). Prior to study participation, written informed consent was obtained from adults or mothers of the children, and assent was obtained from children 7 or more years of age. All procedures at Washington University were approved by the Washington University Institutional Review board (IRB Review Board 01A), and written informed consent was obtained prior to study participation.

### 2.2. Psychophysical Methodology

The same validated psychophysical tools to determine sucrose taste detection thresholds and most preferred sucrose concentration (described in detail below) were used for each age group. From these data, we determined whether there were age-related changes in sweet taste sensitivity and most preferred sucrose concentrations. We also determined whether there were significant associations between taste sensitivity and preference within and across age groups, since both outcomes were measured in the majority of participants.

#### 2.2.1. Sucrose Detection Thresholds

A two-alternative, forced-choice, staircase procedure was used to measure sucrose detection thresholds [15,19]. Sixteen sucrose solutions, ranging from 1000 to 0.1 mM (equivalent to 34% to 0.002% weight/volume) in quarter-log steps, were used. Participants were given pairs of solutions, presented in randomized order: one solution contained sucrose, and the other contained distilled water alone. The first stimulus pair presented to the participant was near the middle of the concentration series (3.2 mM). Participants were asked to taste each sample for 5 s, spit it out, and then point to the sample that had a taste, rinsing once before tasting the second solution and twice between each pair. If they pointed to the water solution, the concentration of sucrose was increased in the next pair. Following two consecutive correct responses, the concentration of sucrose was decreased in the next pair. The testing was complete after four reversals occurred (either from incorrect to correct response, or vice versa). To be considered a reliable threshold and thus included in the final analysis, the four reversals had to meet the following additional criteria recommended by Pribitkin, Rosenthal and Cowart [15]: (1) no more than two dilution steps between two successive reversals, and (2) the series of reversals did not form an ascending pattern (i.e., in which positive and negative reversals are achieved at successively higher concentrations). These additional inclusion criteria likely control for false positives, especially when participants guess and then require a larger number of stimuli presentations to reach criterion [20] and thus ensure a more stable measure of the threshold attained. The mean of the log values of the last four reversals estimated the taste detection thresholds for sucrose in mM (primary outcome).

#### 2.2.2. Most Preferred Sucrose Concentration

The most preferred sucrose concentration was determined via the Monell 2-series, forced-choice tracking procedure that controls for position bias. This procedure has been validated for use across the life span and in children as young as 3 years for the NIH Toolbox [5,14]. Sucrose solutions of varying concentrations were used (160, 240, 380, 610, and 1050 mM, equivalent to 3%, 6%, 12%, 24%, and 36% weight/volume). Participants were presented with pairs of solutions (beginning with the 6% and 24% pair) and asked to taste each sample for approximately 5 s (rinsing once between samples) and then point to which they like better, without instruction on how the stimuli differ. A one-minute interval separated the tasting of each pair, during which participants rinsed their mouth at least twice. Each subsequent pair contained the selected concentration paired with an adjacent stimulus concentration. This pattern continued until the participant chose the same concentration two consecutive times when paired with both a higher and a lower concentration or chose the highest or lowest concentration twice. The entire task was repeated after a 3 min break, with stimulus pairs presented in reverse order to ensure that participants’ responses are not due to choosing whatever comes first or what is in a certain position (e.g., always chooses solution on the right). In the first series, the lower concentration of each pair was always presented first; in the second series, the higher concentration was presented first. This method enables researchers to determine objectively whether the participant is pointing to whatever is presented to the right or left: if this happens, the most preferred concentration in series 1 would be >2 steps apart from the most preferred concentration in series 2, which is an objective criterion for exclusion. The geometric mean (GeoMean) of the most preferred concentrations in the two series estimated the participant’s most preferred sucrose concentration in mM (primary outcome).

### 2.3. Anthropometry

Participants were weighed and measured in light clothing, in duplicate, from which their body mass index (BMI) was calculated. Children and adolescents were placed into BMI categories based on age- and gender-specific BMI z-scores for children and adolescents [21]; for adults, the standard BMI categories were used [22].

### 2.4. Statistical Analyses

The primary outcomes were sucrose taste detection threshold (in mM) and most preferred sucrose concentration (in mM). To determine whether sucrose detection thresholds and preferences differed by age group, we used separate one-way ANOVAs with age group as the between-subjects factor and assessed differences between means using Fisher’s least significance difference tests. In addition, Spearman rank order correlations were calculated to evaluate the strength of the relationship between age (as a continuous variable) and sucrose detection threshold, between age and most preferred sucrose concentration, and between sucrose taste detection threshold and sucrose preference, for each age group separately and all participants combined. To determine whether completion rates differed by age group, we used the Freeman–Halton extension of Fisher’s exact test. With the exception of Fisher’s exact test, that was conducted using Free Statistics Calculator (version 4.0), all other data were analyzed using Statistica software (version 13.3; Tulsa, OK, USA), with a significance criterion set at *p* < 0.05. Data on sucrose detection thresholds were expressed as GeoMean of the last four reversals ± geometric standard errors (GeoSE). GeoSE was calculated as the difference between the GeoMean and the antilog of the arithmetic mean + standard error of the mean (SEM) of transformed data [10^(mean + SEM of the logged data)^—GeoMean], as well as the difference between GeoMean and the antilog of arithmetic mean—SEM of the transformed data [GeoMean—10^(mean − SEM of the logged data)^] (for more details, see [23]). Unless otherwise indicated, data were expressed as the mean and standard deviation (SD).

## 3. Results

### 3.1. Participant Characteristics

Children were, on average, 8.6 (SD: 0.8; range: 7.0–9.9) years of age, and 44.4% were overweight or at risk for overweight. Of the 108 children (41 boys and 67 girls) in the study, 51% were Black, 24% White, and 25% reported to be of other or more than one race. Adolescents were, on average, 13.2 (SD: 2.3; range: 10.0—19.6) years of age, and 45.6% were overweight or at risk for overweight. Of the 172 adolescents (82 boys and 90 girls) in the study, 69% were Black, 13% White, and 18% reported to be of other or more than one race. Adults were, on average, 35.9 (9.4; range: 20.8—67.0) years of age, and 74% were overweight or had obesity. Of the 205 adults (15 men and 190 women) in the study, 53% were Black, 41% White, and 6% reported to be other or more than one race.

Table 1 summarizes the number of individuals per age group with measured taste detection thresholds and preferences, and the percentage of each group that reliably completed each task based on the criteria described in Section 2.2. There were no differences in the percentage of children, adolescents, or adults (>95%) who reliably completed the sucrose preference test; smaller percentages of children and adolescents than adults met criteria for reliable detection thresholds.

### 3.2. Sucrose Detection Thresholds

Sucrose detection thresholds differed significantly among age groups (F_(2,412)_ = 7.1; *p* < 0.001). As shown in Figure 1a, sucrose detection thresholds were higher (i.e., lower sensitivity) in children than either adolescents or adults (*p* < 0.05) and tended to be higher in adolescents than adults (*p* = 0.07). With all groups combined, there was a significant negative correlation between age and sucrose detection thresholds (Spearman *r* = −0.15; *p* < 0.005). To illustrate the age-related differences in sucrose detection thresholds, we estimated the number of 8-ounce (230 mL) glasses of water required per one, 4 g sugar cube to achieve the millimolar detection thresholds for each age group; in all cases, we rounded up. As shown in Figure 1b, the taste detection threshold of children (10.2 mM) was a more concentrated sucrose solution than either adolescents (8.1 mM) or adults (6.8 mM).

### 3.3. Most Preferred Sucrose Concentration

As expected, sucrose preferences differed significantly among age groups (F_(2,375)_ = 18.6; *p* < 0.0001). As shown in Figure 2a, adults preferred significantly lower sucrose concentrations than either children or adolescents (*p* < 0.0001), whose preferences did not differ from each other (*p* = 0.67). With all groups combined, there was a significant negative correlation between age and most preferred sucrose concentration (Spearman *r* = −0.29; *p* < 0.0001). To put this age-related difference in perspective, we estimated the number of sugar cubes that would be required to be dissolved in an 8 ounce glass of water to attain the most preferred concentration for each age group; in all cases, we rounded up the number of cubes. As shown in Figure 2b, the most preferred sucrose concentration of children (600 mM) and adolescents (590 mM) sucrose concentration is equivalent to approximately 12 sugar cubes (each cube contains 4 g sugar) dissolved in 230 mL (8 ounces) of water, whereas the most preferred by adults (400 mM) is equivalent to approximately 8 sugar cubes in the same volume of water.

## 4. Discussion

Age independently associated with taste detection thresholds and preferences for the sweet taste of sucrose. Children had higher sucrose taste detection thresholds, meaning they required higher concentrations of sucrose to detect a taste different from water, compared to adolescents, who in turn required higher concentrations than adults. As shown in Figure 1b, this translates to children requiring a 40% more concentrated sucrose solution to detect a taste, compared to adults.

Children also preferred higher sucrose concentrations than adults, with the changeover occurring during adolescence. As shown in Figure 2b, this translates to approximately 12 sugar cubes dissolved in 230 mL (8 ounces) of water for children but only 8 sugar cubes in the same volume of water for adults. In present times, a typical cola has a sugar concentration of 320 mM, which is similar to the most preferred sucrose concentration in adults.

While the finding that sweet taste preference changes during development is consistent with a large body of research [2,3,4,5,6,7,8], the finding that children are less sensitive to the taste of sucrose than adolescents, who in turn are less sensitive than adults, is rather novel and extends prior research documenting age-related changes from late adolescence to adulthood [9,10,11,24]. That both of these dimensions of sweet taste perception change during development may suggest that age-related differences in sweet taste sensitivity cause children to prefer higher sucrose concentrations; that is, a given sucrose concentration just does not taste as sweet to children as it does to adults. However, that children differed from adolescents in taste detection thresholds but not in sucrose preferences, and the lack of significant relationships between detection thresholds and preferences across age groups, suggests that was not the case. The true test of this hypothesis would be to analyze the relationship between detection and preference within the same individuals as they age.

While taste detection threshold for sucrose and the most preferred sucrose concentration significantly changed with age, there were no significant relationships between the two. When analyzing age groups separately or combined, the most preferred sucrose concentration did not significantly correlate with the lowest concentration of sucrose that the individual detected. These findings are consistent with the lack of relationship between taste detection thresholds and the most preferred concentration of salt in children and adolescents, in a study that used the same methods as the present study [25]. Prior research, using different psychophysical methods to determine taste preferences, also found no relationship between preference and detection thresholds for sucrose in adults [24] or in adolescents [12]. For example, 11- to 15-year-olds who were categorized as high versus low sweet likers, based on their rank ordering by preference of Kool Aid™ solutions that varied in sucrose concentrations, did not differ in sucrose taste detection thresholds [12].

Several explanations, not mutually exclusive, may explain age-related differences in sweet preference and sweet taste detection thresholds and the lack of a relationship between these two dimensions of sweet taste from childhood to adulthood: (1) children’s inability to understand the task or difficulty sustaining attention, (2) age-related changes during adolescence resulting from distinct ontogenetic trajectories, and/or (3) heightened preference for sweets in children and adolescents reflecting greater physiological needs.

First, the age-related differences in both taste detection and preferences may have been due to children’s inability to understand the task or difficulty sustaining attention. However, both psychophysical tools used in the present study eliminated the need for verbal responses and controlled for position bias, which allowed for the objective determination of whether the participant comprehended the task or responded at random. The ability of children and adolescents to comprehend and complete the preference task was similar to that of adults. Further, their most preferred sucrose concentrations were not higher than adults simply because they repeatedly choose the highest sucrose concentration (1050 mM, 36% weight/volume) offered. While the taste detection threshold task was relatively more difficult for the younger participants than for adults, both psychophysical tasks had inclusion criterion established a priori and recommended clinically [15] and, thus, it is unlikely that the age-related changes are due to differences in task performance.

Second, age-related changes in sucrose preferences and detection thresholds that occur during adolescence may result from distinct ontogenetic trajectories with different underlying mechanisms. The appetite or preferences for sweet taste may be consequences of central changes in the activity and morphology of the brain reward system [26,27]. Using the same psychophysical methods we used here to measure sucrose preference, we found that the binding potential of dopamine receptors in striatum, a brain area that encodes reward value, decreased with age and also predicted, independently of age, the most preferred sucrose concentration in healthy young adults [28].

On the other hand, developmental changes in detection thresholds may be secondary to peripheral changes in the anatomy of the oral cavity and the composition of saliva. It has been hypothesized that, as the surface of the tongue increases with body weight and with larger surface in primates, there would be greater numbers of taste buds to elicit a greater signal, thus resulting in increased sensitivity [29]. In addition, changes occur during adolescence in the protein content and protein activity of saliva [30,31]. For example, the activity of salivary amylase, an enzyme that catalyzes the breakdown of starch into sugar as a first step in the digestion process, increases from undetectable in newborns to adult levels by adolescence [30]. Whether there are also age-related changes in alpha-glucosidase digestive enzymes expressed in taste buds remains unknown, but if so, we hypothesize that such changes in these enzymes that contribute to taste cell responses to disaccharides [32] could underlie age-related changes in sucrose detection thresholds.

Third, the heightened preference for sweets in children and adolescents may, in part, reflect greater energetic needs due to growth [3,33,34]. In support of this hypothesis, the level of sweetness most preferred has been shown to be significantly related to the height of children and urinary levels of N-terminal telopeptide of type 1 collagen, a biomarker for bone resorption and growth [12,33]. Remarkably, the developmental-related change in sweet taste preference is not unique to humans since it has been observed in other mammalian orders, such as rodents [34].

The attraction that youth have for “sweetness” reflects their inborn biology. It is believed that evolutionary pressures shaped the taste of foods initially preferred and rejected. In an environment with limited nutrients and abundant poisonous plants, sensory systems evolved to detect and reject perceptions that specified potential poisons that taste bitter [35] and to prefer perceptions that specify crucial nutrients such as the once rare carbohydrate (energy)-rich plants that taste sweet [29].

Whether age-related changes in taste sensitivity have selective advantages remains speculative. Unlike sweet taste, where sensitivity increases from childhood to young adulthood, the reverse has been observed for some bitter substances, a taste signal for potential poisons. For some bitter agents, children are more sensitive in detecting taste (lower detection thresholds) than adults, with the changeover occurring in adolescence [36]. Although we have shown that 0.6 M sucrose solution can suppress bitterness in a variety of taste mixtures in both children and adults [37], it remains unknown whether the relatively lower sweet detection thresholds in children modulate the efficacy of sugars in masking bitterness, which in turn would increase the likelihood that they would detect toxins in foods that contain both bitter compounds and small amounts of sugars.

The present findings must be interpreted considering some limitations. First, the cross-sectional study design does not allow determination of causality. Second, the nature of this secondary data analysis resulted in a sample of adults, the majority of whom were women who had overweight or obesity. However, it is unlikely that the sex and BMI imbalance among the adult group were factors for a number of reasons. First, both sucrose detection thresholds and preferences were within the range of previously published work [2,3,4,5,6,7,8,13,15,38,39,40,41]. Second, prior research found no differences in either measure between men and women [3,9,13,41] or adults with normal weight and those with obesity [28,42]. Third, the absence of dietary intake data precludes the evaluation of how age-related changes in dietary intake [43] contribute to changes in taste sensitivity and preference for sweet tastes. Future longitudinal research to determine how dietary intake of added sweeteners (with or without calories) impacts these two dimensions of the sweet taste system over time is warranted.

## 5. Conclusions

The present findings suggest that information on the sensitivity of the sweet taste should not be used as proxy for sweet taste hedonics and vice versa and that these two dimensions of sweetness perception undergo distinct developmental trajectories from childhood to adulthood. Taken together, this evidence suggests that children’s heightened preference for sweet taste and their proclivity for sweet-tasting foods and beverages cannot be explained simply by a reduced taste sensitivity to sweet taste.

## Figures and Tables

**Figure 1 nutrients-12-01918-f001:**
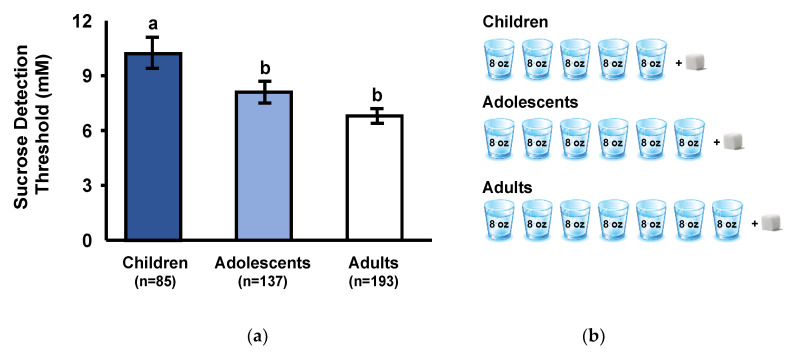
(**a**) Sucrose taste detection thresholds: concentration of sucrose (mM) at which children, adolescents, and adults detected a taste different from water. There was a significant effect of age group (*p* < 0.001). Different subscripts represent groups that are significantly different from each other at *p* < 0.05. Values are presented as GeoMeans ± GeoSE (see Section 2.4 for explanation). (**b**) To illustrate the age-related differences in sucrose detection thresholds, we converted millimolar concentrations of each age group to the number of 8-ounce (230 mL) glasses of water required to dilute one 4 g sugar cube, in all cases we rounded up.

**Figure 2 nutrients-12-01918-f002:**
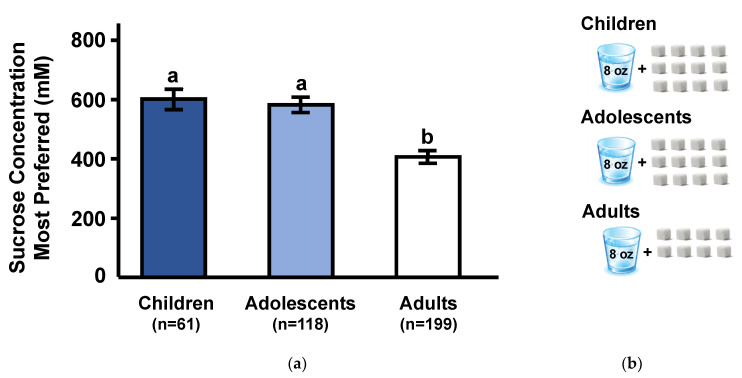
(**a**) Sucrose concentration (mM) most preferred among children, adolescents, and adults. There was a significant effect of age group (*p* < 0.001). Different subscripts represent groups that are significantly different from each other at *p* < 0.05. Values are means ± SEM (see Section 2.4 for explanation). (**b**) To illustrate the age-related differences in most preferred sucrose concentration, we converted millimolar concentrations of each age group to number of 4 g sugar cubes required per one 8-ounce (230 mL) glass of water, in all cases we rounded up.

**Table 1 nutrients-12-01918-t001:** Task Performance by Age Group.

	Age Group			
	Children	Adolescents	Adults	*p*-Value
**Number Enrolled**	108	172	205	
**Sucrose Taste Detection Threshold**				<0.001
Tested ^a^, *n*	106	165	203	
Included, understood, and completed *n* (%)	85 (80%)	137 (83%)	193 (95%)	
Excluded ^b^, *n* (%)	21 (20%)	28 (17%)	10 (5%)	
**Sucrose taste preference**				0.14
Tested, *n*	64	122	201	
Included, understood, and completed, *n* (%)	61 (95%)	118 (97%)	199 (99%)	
Excluded ^b^, *n* (%)	3 (5%)	4 (3%)	2 (1%)	

^a^ Number of individuals do not sum to total enrolled due to dropouts. ^b^ Individuals were excluded if they responded at random, did not complete the task, or did not reach criteria (see Section 2.2 for details).

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
