# Peer review of "Relationship between Sucrose Taste Detection Thresholds and Preferences in Children, Adolescents, and Adults"

_nutrients, 2020, doi:10.3390/nu12071918_

Round 1

Reviewer 1 Report

This study examines the differences in sweet taste sensitivity and preference in children, adolescents and adults using the prototypical sweet taste stimulus, sucrose. The study is well designed, and the manuscript is well written. The finding that adolescents have higher sweet taste sensitivity than children yet have comparable sweet taste preference threshold is very interesting. This indicates that the switch to preference for lower sugar concentrations happens after this stage and will hopefully stimulate interest in longitudinal studies of sweet taste sensitivity, preference and eating habits. Curiously, most adult participants in this study were women. Gender differences in sweet taste sensitivity is known and readers will benefit from including this in the discussion section. Similarly, relation between obesity and sweet taste sensitivity might also be helpful (most participants in this study are obese) although this is already documented in literature.

A few specific grammatical errors are listed below.

Line 59: Do the authors mean “To address these aims…” ?

Line 219: Do the authors mean “Children also preferred significantly…” ?

Author Response

We would like to thank the Reviewers for their thoughtful and constructive evaluation of our manuscript, entitled “Relationship between sucrose taste detection thresholds and preferences in children, adolescents, and adults” and appreciate the opportunity to submit a revised version. Below are our point-by-point responses to the Reviewers’ comments.

Reviewer 1

Comment 1. This study examines the differences in sweet taste sensitivity and preference in children, adolescents and adults using the prototypical sweet taste stimulus, sucrose. The study is well designed, and the manuscript is well written. The finding that adolescents have higher sweet taste sensitivity than children yet have comparable sweet taste preference threshold is very interesting. This indicates that the switch to preference for lower sugar concentrations happens after this stage and will hopefully stimulate interest in longitudinal studies of sweet taste sensitivity, preference and eating habits. Curiously, most adult participants in this study were women. Gender differences in sweet taste sensitivity is known and readers will benefit from including this in the discussion section. Similarly, relation between obesity and sweet taste sensitivity might also be helpful (most participants in this study are obese) although this is already documented in literature.

 Reply, Comment 1.  We thank the Reviewer for his/her compliments and also hope the research will stimulate longitudinal research in the area. Per the Reviewer’s recommendation, we now include a discussion on potential associations between sweet taste sensitivity/preferences and sex or obesity.

The new text reads:” The present findings must be interpreted considering some limitations. First, the cross-sectional study design does not allow determination of causality. Second, the nature of this secondary data analysis resulted in a sample of adults, the majority of whom were women who had overweight or obesity. However, it is unlikely that the sex and BMI imbalance among the adult group were factors for a number of reasons. First, both sucrose detection thresholds and preferences were within the range of previously published work [2-8,13, 15,38-41]. Second, prior research found no differences in either measure between men and women [3, 9,13, 41] or adults with normal weight and those with obesity [28, 42].

Comment 2.  A few specific grammatical errors are listed below.

Reply, Comment 2: We edited the manuscript throughout and corrected all grammatical errors.

 Comment 3. Line 59: Do the authors mean “To address these aims…” ?

Reply, Comment 3.  Yes, the sentence has been corrected.

 Comment 4.  Line 219: Do the authors mean “Children also preferred significantly…” ?

Reply, Comment 4. Yes, the sentence has been corrected.

Reviewer 2 Report

This manuscript describes the relationship between sucrose detection thresholds and sweetness preferences, for children (<10 years), adolescents (10-19 years) and adults (>19 years). The work is well designed and implemented. The findings are thoroughly discussed relative to the literature.

The authors did point out that: the lack of dietary intake information prevented the assessment of how food-intake influenced sugar sensitivity and sweetness preferences; however, they did not comment on the fact that the adults in the study were particularly overweight (74% overweight) compared to the children/adolescents (44.4-45.6% overweight). I question whether this was a ‘representative sample’ for the adults in the US, and if such a rate would likely influence sugar sensitivity and/or sweetness preference.

Otherwise, the manuscript contains only a few other oversights that are relatively minor in nature:

  • The phrase “the concentration most preferred…” and “the concentration of sucrose most preferred” that occur throughout the manuscript seem to have a rather awkward syntax. The authors might consider replacing these phrases with “the most preferred sucrose concentration”.
  • Line 30: The word “bundle” is an unusual choice of words - replace.
  • Line 42: The word “most” in this sentence should be eliminated.
  • Lines 51, 52: Two sentences back-to-back start with the conjunction “however” – revise.
  • Lines 75-76: The last part of the sentence could be improved by using parallel sentence structure “…were either normal weight or with obesity.”
  • Line 90: Replace “hedonics” with “hedonic evaluations”
  • Lines 89-93: These five lines have very complex wording - improve/rewrite.
  • Line 96: The concentrations used for the hedonic evaluations were converted to percentages (lines 117-118). It would be useful to provide the equivalent percentages for the threshold evaluations too.
  • Line 171: This table has alignment problems in the first column, making it difficult to read.
  • Table 1: This table reports the p-values for the completion rates associated with the three groups of participants. However, this statistical analysis was not described in the materials and methods.
  • Line 181: The numerical value “1” in this sentence should be converted to text.
  • Lines 208-209: This sentence is incomplete or missing some of the necessary details.
  • Line 219: “Children also most preferred was significantly higher concentrations of sucrose...”. This sentence doesn’t make sense - revise.
  • Lines 227-230: The structure and syntax of this sentence is particularly awkward and difficult to understand – rewrite.
  • Lines 255-257. This sentence contains a double negative.

Author Response

We would like to thank the Reviewers for their thoughtful and constructive evaluation of our manuscript, entitled “Relationship between sucrose taste detection thresholds and preferences in children, adolescents, and adults” and appreciate the opportunity to submit a revised version. Below are our point-by-point responses to the Reviewers’ comments.

 Reviewer 2

 Comment 1. This manuscript describes the relationship between sucrose detection thresholds and sweetness preferences, for children (<10 years), adolescents (10-19 years) and adults (>19 years). The work is well designed and implemented. The findings are thoroughly discussed relative to the literature.

The authors did point out that: the lack of dietary intake information prevented the assessment of how food-intake influenced sugar sensitivity and sweetness preferences; however, they did not comment on the fact that the adults in the study were particularly overweight (74% overweight) compared to the children/adolescents (44.4-45.6% overweight). I question whether this was a ‘representative sample’ for the adults in the US, and if such a rate would likely influence sugar sensitivity and/or sweetness preference.

 Reply, Comment 1.  We have modified the discussion to include that one of the limitations of the study is that the majority of adults were women and were either normal weight or with obesity. However, it is unlikely that the sex and BMI imbalance among the adult group were factors. First, both sucrose detection thresholds and preferences were within the range of previously published work [2-8,13, 15,3 8-41]. Second, prior research found no differences in either measure between men and women [3, 9,13, 41] or adults with normal weight and those with obesity [28, 42].

Comment 2. The phrase “the concentration most preferred…” and “the concentration of sucrose most preferred” that occur throughout the manuscript seem to have a rather awkward syntax. The authors might consider replacing these phrases with “the most preferred sucrose concentration”.

 Reply, Comment 2. Per the Reviewer’s suggestion, we have use the phrase with “the most preferred sucrose concentration” throughout the text.

 Comment 3. Line 30: The word “bundle” is an unusual choice of words - replace.

Reply, Comment 3. Per the Reviewer’s suggestion we edited that sentence and replace bundle by array of attributes.

 Comment 4. Line 42: The word “most” in this sentence should be eliminated.

Reply, Comment 4.  The word has been eliminated from the sentence.

 Comment 5. Lines 51, 52: Two sentences back-to-back start with the conjunction “however” – revise.

Reply, Comment 5. The sentences have been revised. 

Comment 6. Lines 75-76: The last part of the sentence could be improved by using parallel sentence structure “…were either normal weight or with obesity.”

Reply, Comment 6. The sentence has been edited.

 Comment 7. Line 90: Replace “hedonics” with “hedonic evaluations”; Lines 89-93: These five lines have very complex wording - improve/rewrite.

Reply, Comment 7. Per the Reviewer’s suggestion we rewrite those lines. Which now read:

The same validated psychophysical tools to determine sucrose taste detection thresholds and most preferred sucrose concentration (described in detail below) were used for each age groups. From these data, we determined whether there were age-related changes in sweet taste sensitivity and hedonic value.  Additionally, because both outcomes were measured in the majority of participants, we also determined whether sucrose taste detection thresholds were associated with most preferred sucrose concentrations in each age group and whether such relationships changed with age.

 Comment 8, Line 96: The concentrations used for the hedonic evaluations were converted to percentages (lines 117-118). It would be useful to provide the equivalent percentages for the threshold evaluations too.

Reply, Comment 8. The equivalent percentages for the threshold evaluations are now provided in the text. 

 Comment 9, Line 171: This table has alignment problems in the first column, making it difficult to read.

Reply. We agree with the Reviewer but the original submission in which the first column was left justified was reformatted by the Journal before sending out for review. The Journal instructions state that the text of each column  in the table should be  “centered”.

Comment 10, Table 1: This table reports the p-values for the completion rates associated with the three groups of participants. However, this statistical analysis was not described in the materials and methods.

Reply. We apologize for that oversight. We now clarify in the statistical analysis section:

“To determine whether completion rates for each taste test differed by age group, we used   the Freeman-Halton extension of Fisher exact test. With the exception of Fisher exact test, that was conducted using Free Statistics Calculator (version 4.0), all other data were analyzed using Statistica software (version 13.3; Tulsa, OK), with a significance criterion set at p<0.05.”

 Comment 11, Line 181: The numerical value “1” in this sentence should be converted to text.

Reply, Comment 11. The numerical value has been replaced by the word ‘one’.

 Comment 12, Lines 208-209: This sentence is incomplete or missing some of the necessary details.

Reply, Comment 12. The sentence now reads: “Different subscripts represent groups that are significantly different from each other at p<0.05.”

 Comment 13, Line 219: “Children also most preferred was significantly higher concentrations of sucrose...”. This sentence doesn’t make sense - revise.

Reply, Comment 13. The sentence now reads: “Children also preferred higher sucrose concentrations than did adults with the changeover occurring during adolescence”.

 Comment 14, Lines 227-230: The structure and syntax of this sentence is particularly awkward and difficult to understand – rewrite.

Reply, Comment 14. We edited the sentence and we hope is now clearer. “While taste detection threshold for sucrose and the most preferred sucrose concentration significantly changed with age, there were no significant relationships between the two. When analyzing age groups separately or combined, the most preferred sucrose concentration did not significantly correlate with the lowest concentration of sucrose that the individual detected.”

 Comment 15, Lines 255-257. This sentence contains a double negative.

Reply, Comment 15.  We rewrote the sentence to remove the double negative. The sentence now reads: “The ability of children and adolescents to comprehend and complete the preference task was similar to that of adults. Further, their most preferred sucrose concentrations were not higher than adults simply because they repeatedly choose the highest sucrose concentration (1050 mM, 36% weight/volume) offered”.